# Controlling Grain Sizes of 42CrMo Steel by Pre-Stress Hardening Grinding

**DOI:** 10.3390/ma12193124

**Published:** 2019-09-25

**Authors:** Yushi Wang, Shichao Xiu, Shengnan Zhang

**Affiliations:** 1Mechanical and Electrical Engineering Institute, Shenyang Aerospace University, Shenyang 110136, China; 2Department of Mechanical Engineering and Automated, Northeastern University, Shenyang 110819, China; shchxiu@mail.neu.edu.cn (S.X.); snzhang@stumail.neu.edu.cn (S.Z.)

**Keywords:** dynamic recrystallization, PSHG, microstructure, grain size, modeling, mechanical property

## Abstract

The dynamic recrystallization behavior of 42CrMo steel during the pre-stress hardening grinding (PSHG) process was investigated at temperatures ranging from 850–1150 °C and pre-stress from 0 MPa to 167 MPa. A coupled grain size model considering different grinding conditions was constructed to research the grinding process. Microstructure analyses showed that the hardening layer exhibits the typical features of dynamic recrystallization (DRX), and the evolution process of microstructure and grain size can be predicted properly by the model. The volume fraction of DRX grains increases with increasing pre-stress and grinding temperature. The critical condition for DRX grains occurring is that with a grinding depth of 150 µm, pre-stress is larger than 67 MPa, while most of the DRX grains occurred when pre-stress is larger than 100 MPa. Furthermore, the relationship between pre-stress and flow stress has been derived. The result shows that flow stress shows a linearly increasing trend, with the increase of pre-stress at the stage of lower strain.

## 1. Introduction

Pre-stress hardening grinding (PSHG) is a compound grinding technology which can not only harden a workpiece surface, but also control residual stress [1]. The results of experiments indicated that the workpiece surface can obtain a hardening layer with excellent surface integrity after the PSHG process, and therefore has an advantage over traditional grinding and heat treatment [2,3,4]. The metallic materials with lower stacking fault energy generally include the stage of work hardening (WH) and dynamic softening (induced by dynamic recrystallization (DRX)) during hot deformation. Therefore, microstructure evolution in the grinding deformation region, possibly under the influence of temperature, pre-stress and grinding force, creates tiny grains, which are called DRX grains. There have been varied theoretical, mathematical, and numerical methods to model DRX behavior, but most of them are conducted based on hot plastic deformation process. Much of the existing literature [5,6,7] shows that strengthening and grain refinement are the common characteristics of recrystallized microstructures, and the grains are equiaxed in shape. Therefore, it has significant importance for investigating the effect of DRX behavior on microstructure evolution during PSHG technology.

The studied material in this paper is 42CrMo, which is a medium carbon and ultra-high strength steel. Due to its excellent hardenability and higher fatigue limit after hardening and tempering, surface hardening is usually used as heat treatment for this steel. A growing number of extensive researches have been conducted on microstructural evolution for various alloys in recent years. By considering the effect of grain size on the degree of recrystallization, Zerilli and Armstrong [8] presented a constitutive relation into which the effects of strain hardening and high temperature softening based on true stress–strain analysis has been incorporated. The initiation and evolution of DRX were investigated through hot compression experiments [9,10,11], and systematic analysis coupling effect between dynamic recrystallization (DRX) behavior and flow stress. Li et al. [12] established a Poliak–Jonas (P–J) model to calculate critical strain and kinetics of DRX through quantitative analysis of the regression of stress–strain data obtained in the experiment under different conditions. Poliak and Jonas [13] identified the onset of DRX based on a one-parameter approach. The critical point of DRX corresponds to the inflection point on work-hardening curves, and the initiation of DRX is governed by both extrinsic incentive and thermal activation energy. Chen et al. [14] investigated the surface gradient microstructure of a Ni-based superalloy induced by high-speed machining and estimated the multiscale texture of undeformed coarse grain region. The hierarchical model is employed to describe the dislocation substructure. In addition, the nucleation mechanism of DRX is well explained by bulging of initial grain boundaries in recent studies [15,16]. Through means of isothermal compression tests, Zhang et al. [17] investigated whether the effect of strain rate on the microstructure evolution for nickel-based superalloy, high dislocation density, and adiabatic temperature rise would stimulate nucleation of recrystallization. In addition, the relationship between dynamic strain and nucleation of DRX has been verified. In recent years, more scholars have paid attention to the microstructure evolution and its influence on physical and mechanical properties of materials. However, most of the research concentrated on hot deformation of metallic materials, and there are only a few systemic investigations into DRX evolution during compound grinding. 

DRX grain occurring should be satisfied with sufficient thermal activation energy, critical strain, lower strain rate, and higher dislocation density. Utilizing this characteristic, this paper proposes that DRX behavior occurs under specific grinding conditions, and quantitatively analyzes the relationship between pre-stress and grain size. 

## 2. Experiments

### 2.1. Materials, Specimens, and Experimental Equipment

The experimental material was 42CrMo high-strength steel, the composition of which is presented in Table 1. The specimens were machined into a specified shape with a machining section measuring 50 × 22 × 9mm (l × b × h). The PSHG tests were conducted on a M250 surface grinder (Junfeng machine, Yancheng, China), which is shown in Figure 1a, and the formation process of DRX grains during PSHG is shown in Figure 1b,c. During the PSHG process, the 42CrMo steel underwent high temperature and pre-stress, which induced dislocation density to increase to a high level. Besides this, a certain thickness of plastic deformation layer formed in the workpiece surface with the increase of pre-stress. Once the deformation reached a critical value, it resulted in higher deformation energy stored near the grain boundaries and deformation bands. Therefore, the driving force is considered primary, provided by pre-stress and grinding temperature. The newly formed DRX grains will replace the initials, and the average grain size is refined after a series of microstructural evolution processes. The main technological parameters are the following: main motor power, 3.04 KW; highest rotating speed, 2850 r/min; workbench area, 540 mm × 250 mm; wheel size, 200 mm × 20 mm × 32 mm. The white alumina abrasives grinding wheel was used in the experiments. 

### 2.2. Experimental Procedures

In order to consider the effect of pre-stress on microstructure evolution, different pre-stresses were applied to both sides of the workpieces with a pre-stress applying device before the grinding tests. In addition, the different temperatures were obtained by controlling grinding depth. This process produces a massive amount of heat, which mainly transfers to the workpiece surface. Due to the effect of pre-stress and grinding temperature, the material in the workpiece surface will produce thermal deformation. Therefore, with properties of low-stacking fault energies, the DRX grain will occur on workpiece surface. Then, the workpieces were cooled in the air to room temperature. The detail parameters are listed in Table 2. Finally, the workpieces were sliced into metallographic specimens and corroded with saturated picric acid. In addition, the average grain sizes were measured using the linear intercept method [18].

### 2.3. Theory of Developed Models

DRX is considered to be a re-growth of grains when the stored energy reaches the critical value during hot deformation. Cellular automata (CA) models are well suited for describing the evolution process of complex systems such as flow stress, DRX fraction, pre-stress, and grain size. To visualize the evolution process, the system variables are regarded as discrete cells and certain transformational rules were used to describe the states between the cells and their Moore neighborhood. The local status values are limited and varied in time and space, which can be calculated by transition rules. The developed CA model established in this study consists of four modules: flow stress module, thermal activation energy module, recrystallization fraction module, and grain size module. Hence, the developed model can be expressed as
(1)$$D_{drx}=\alpha_{1}F\left(\sigma \right)\beta_{1}\ln{\left(1-X_{DRX}\right)}^{m_{1}}\gamma_{1}Z^{n_{1}}$$
where F(σ) represents flow stress (MP) in different conditions, X*_DRX_* represents volume fraction of DRX, *Z* is the well-established Zener–Hollomon parameter, $\alpha_{1}$, $\beta_{1}$, $\gamma_{1}$, $m_{1}$ and $n_{1}$ represent material constants, $D_{drx}$ represents grain size of DRX.

## 3. Results and Discussion

### 3.1. Phase Transition and Dynamics

As the workpieces were machined with a larger grinding depth during PSHG, the temperature on the surface can reach the conditions for austenite transformation. Therefore, the martensite microstructure can be observed after cooling. As shown in Figure 2, the white and needle-like structures are martensite microstructure, which is stretched by pre-stress. In conclusion, microstructure is not only associated with transition temperature but also internal stress, induced by pre-stress. Meanwhile, the grain size is also affected by pre-stress. This meant that the PSHG technology is able to harden the workpiece surface and control grain size. Therefore, it is necessary to investigate the variation of stress internal microstructure.

Based on the effects of temperature and strain rate, the kinetic models of DRX are established as following.

The Arrhenius equation is commonly accepted to better describe the parameters affected by the DRX process [19,20].
(2)$$\dot{\varepsilon }=AF\left(\sigma \right)exp\left(-Q/RT\right)$$
(3)$$F\left(\sigma \right)=\{\begin{array}{ll} \sigma^{n} & \alpha \sigma <0.8\\ \exp\left(\beta \alpha \right) & \alpha \sigma >1.2\\ {\left[\sinh\left(\alpha \sigma \right)\right]}^{n} & for\;all\;\sigma \end{array}$$
where *σ* represents flow stress (MP), $\dot{\varepsilon \;}$ represents strain rate ($s^{-1}$), T represents absolute temperature (K), *Q* represents thermal activation energy, n, *β*, A, and *α* represent material constants, and *R* represents gas constant (8.314 J mol^−1^ K^−1^), among which stress exponent α can calculate as α = *β*/n.

Figure 3a–d show the typical stress–strain curves of 42CrMo steel which are obtained under different deformation temperatures (1123 K,1223 K,1323 K,1423 K) and strain rates (0.01 s^−1^,0.1 s^−1^,1 s^−1^). Obviously, it can be found from the curves that the variation in stress is very sensitive to deformation temperature and strain rate. Comparing these curves, the flow–stress value decreases with rising deformation temperature and reducing strain rate at the same strain. It lies primarily in the fact that higher temperature can provide higher mobility for grain boundaries, while at the same time there is enough time to nuclear and grow for recrystallized grains under lower strain rate, which results in more DRX grains occurring and dislocation annihilation. In addition, it can be found that the process of stress variation consists of three periods. At the initial stage, work hardening (WH) is primarily a mechanism by which flow stress presents a rapid increase at this stage. For the second stage, when deformation exceeds peak strain, the obvious stress decrease could be observed. The cause lies in the fact that higher deformation energy drives the piled-up dislocations consumed to form DRX nucleation, and this process is called dynamic softening. At the third stage, the flow–stress curves decrease gradually to a stable value under the interaction between work hardening and DRX softening.

When the stress level is lower (ασ<0.8), substituting $F\;\left(\sigma \right)=\sigma^{n}$ into Equation (2) and taking the nature logarithm on both sides. Then $n=\mathrm{dln}\dot{\varepsilon }/\mathrm{dln}\sigma .\;$ According to the research in References [9,10,11], within a strain range from 0.08 mm to 0.18 mm for 42CrMo high-strength steel, the stresses increased almost linearly and the effect of temperature on the ratios was very small. Thus, in order to evaluate the constant n, true strain ε = 0.12 mm was chosen in this study. In addition, when the stress level was higher (ασ>1.2), the identical manner was used to obtain material constant $\beta =\mathrm{dln}\dot{\varepsilon }/d\sigma$. Figure 4 shows the plots of $ln\dot{\varepsilon }-ln\sigma$ and $ln\dot{\varepsilon }-\sigma$ under different deformation temperatures. The average value of slope rate of fitting curves is the desired value. Thus, the material constant n, β, and α can be obtained with the values of 9.0296, 0.0815, and 0.00903, respectively.

It is well accepted that *Z* parameter can be quantitatively described with strain rate and deformation temperature:(4)$$Z=\dot{\varepsilon }\exp\left(Q/RT\right)$$

Taking logarithm on both sides of Equation (2), the equation converted into:(5)$$\ln\dot{\varepsilon }=\mathrm{lnA}+n\left[\ln\sinh\left(\alpha \sigma \right)\right]-Q/\mathrm{RT}$$

It can be found from Equation (5) that it shows a linear relationship between lnsinh(ασ) and 1/T. Thus, *Q* is calculated as 551.63 kJmol^−1^. Substituting the calculated material constants into Equation (5), constant A can be evaluated as 6.8347 × 1022 s^−1^.

Therefore, the mathematical expression for flow stress at different conditions is derived as following:(6)$$\sigma =110.7420ln\left\{{\left[\frac{Z}{6.8347\times 10^{22}}\right]}^{1/9.0296}+{\left\{{\left[\frac{Z}{6.8347\times 10^{22}}\right]}^{2/9.0296}+1\right\}}^{1/2}\right\}$$

### 3.2. Modeling of Dynamic Recrystallization (DRX) Kinetics

The hardening rate curves (θ−σ) shown in Figure 5 which are often used to determine whether or not DRX behavior occurs, moreover, critical strain ($\varepsilon_{c}$), critical stress ($\sigma_{c}$), peak stress ($\sigma_{P}$), and peak strain ($\varepsilon_{p}$) are easy to estimate from the curves. Research suggests that the onset of DRX is considered to correspond to the inflection point on hardening rate curves, which means the minimum of first derivative. This identification method has been verified by extensive research, so in this study it was used to reveal whether DRX occurs in the PSHG process. The hardening rate curves are plotted under different strain rates of 0.01 s^−1^ and 0.1 s^−1^, grinding temperatures range from 1123 K to 1423 K. The DRX process occurs under the thermo–stress coupled process, and strain and thermal activation energy are both essential. The inflection point of curve is considered as $\sigma_{c}$ and its value can be calculated as $\partial^{2}\theta /\partial \sigma^{2}$, as well as σ_p_ is considered to correspond to the point at which *θ* equals 0. In addition, the corresponding strains $\varepsilon_{c}$ and $\varepsilon_{p}$ are easy to obtain from the stress–strain curves. Table 3 presents the values of $\varepsilon_{c}$ and $\varepsilon_{p}$ at different conditions.

After the complex DRX process, the average grain size in surface layer is refined, and the mechanical property is also improved. The average grain size is closely connected with the DRX volume fraction. The Avrami model is widely accepted to express the relationship between DRX volume fraction and strain. The expression is as follows [21].
(7)$$X_{\mathrm{DRX}}=1-\exp\left(-K_{D}{\left(\frac{\varepsilon -\varepsilon_{c}}{\varepsilon_{p}}\right)}^{n_{D}}\right)\;\;\;\;\;\;\;\;\;\;\;\;\;\;\;\;\;\;\;\;\;\;\;\;\left(\varepsilon >\varepsilon_{c}\right)$$
where X_DRX_ represent volume fraction of DRX, $\varepsilon_{c}\;\mathrm{represents}$ critical strain, $\varepsilon_{p}\;\mathrm{represent}$ peak strain, K_D_ and $n_{D}$
represent material constants.

The DRX volume fraction (X_DRX_) can be quantitatively calculated by stress–strain curve analysis [22,23] This method is commonly used to determine X_DRX_, which can be expressed as
(8)$$X_{\mathrm{DRX}}=\frac{\sigma_{\mathrm{rev}}-\sigma }{\sigma_{S}-\sigma_{\mathrm{SS}}}$$
where $\;\sigma_{\mathrm{ss}}\;\mathrm{represents}\;\mathrm{steady}\;\mathrm{stress}\;\left(\mathrm{MPa}\right),\;\mathrm{and}\;\sigma_{s}$ represents saturation stress (MPa). $\sigma_{\mathrm{rev}}$ represents dynamic recovery (DRV) occurs mainly at this stage. Additionally, the value of $\sigma_{\mathrm{rev}}-\sigma$ can be estimated by utilizing the value of $\sigma_{p}-\sigma \;$. Thus, the X_DRX_ can be expressed as
(9)$$X_{\mathrm{DRX}}=\frac{\sigma_{p}-\sigma }{\sigma_{S}-\sigma_{\mathrm{SS}}}$$

Generally, there is a power function among peak strain ($\varepsilon_{p}$), critical strain ($\varepsilon_{c}$), and *Z* parameter [24,25]. The relationship between $\ln\varepsilon_{p}-\ln\left(Z\right)$ and $\ln\varepsilon_{c}-\ln\left(Z\right)$ is plotted in Figure 6. By substituting the values of $\;\varepsilon_{p}\;,\;\varepsilon_{c},$ and X_DRX_ into Equation (7), the material constants K_D_ and n_D_ can be calculated as 0.4789 and −0.2903, respectively.

Through the above analysis, the DRX kinetic models can be summarized as:(10)$$\{\begin{array}{l} X_{\mathrm{DRX}}=1-\exp\left(-0.4789{\left(\frac{\varepsilon -\varepsilon_{c}}{\varepsilon_{p}}\right)}^{0.2903}\right)\;\;\;\;\;\;\;\;\;\;\;\;\;\;\;\;\;\;\;\;\;\;\;\;\;\;\left(\varepsilon >\varepsilon_{c}\right)\\ \;\varepsilon_{p}=0.0234\times Z^{0.0466}\\ \;\varepsilon_{c}=0.0189\times Z^{0.0366} \end{array}$$

### 3.3. The Relationship between Pre-stress and Grain Size

Due to workpiece deformation mostly depending on pre-stress in PSHG, the interaction between work hardening and DRX softening on microstructure evolution is constant under a certain pre-stress. Experimental study shows that the flow stress is closely interrelated with pre-stress, and their relationship at deformation temperatures of 1323 K and 1423 K is shown in Figure 7a,b. It can be found that $\sigma -\sigma_{p}$ curves include two stages. At the first stage, flow stress shows a linearly increasing trend with increasing pre-stress. For the second stage, the flow stress shows a decrease first and then restores a trend as pre-stress increases. In addition, from the figure the $\sigma -\sigma_{p}$ curves are sensitive to strain rate, but temperature has almost no effect on it. The stress dependence model up to saturated flow stress can be calculated from the first stage. Though linear fitting analysis, the relationship between flow stress and pre-stress can be given as follows: (11)$$\left\{ \begin{array}{l} \sigma_{p}=8.6071\sigma -7.5490\\ \sigma_{p}=9.1802\sigma -20.1356 \end{array}\right.$$

Research suggests that there is a power function between flow–stress and DRX grain size, which can be defined as follows [26].
(12)$$\sigma /G=AD_{drx}^{n}$$

Where *A* and n are constant, G is shear modulus of material, and $D_{drx}$ is the DRX grain size.

In order to further determine the constant, the universal relationship is developed for average grain sizes and stresses, which are suitable for most materials [27].
(13)$$1<\frac{\sigma }{G}{\left(\frac{D}{b}\right)}^{2/3}<10$$

As shown in Figure 7c,d, the power function curves are obtained though fitting the experimental results at the temperature of 1323 K and strain rate of 0.01 s^−1^ and 0.1 s^−1^. It can be found that the average grain size decreases as flow stress increases. The DRX grain size is not only sensitive to pre-stress, but also sensitive to strain rate and temperature. At a specific temperature of 1323 K, the grain size models are obtained by nonlinear fitting method, which can be expressed as
(14)$$\left\{ \begin{array}{l} \sigma /G=0.0543D_{drx}^{-0.9049}\\ \sigma /G=0.0915D_{drx}^{-1.0003} \end{array}\right.$$

By substituting Equation (11) into Equation (14), the newly constructed numerical model of DRX grain size in specific conditions can be expressed as Equation (15), by which the constructed model can quantitatively predict grain size by controlling pre-stress.
(15)$$\left\{ \begin{array}{l} \sigma_{p}/G=0.4674D_{drx}^{-0.9049}-7.5490\\ \sigma_{p}/G=0.7875D_{drx}^{-1.0003}-7.5490 \end{array}\right.$$

For a more intuitive display of the established models, simulations are performed for 42CrMo steel at different pre-stress, but keep the temperature and strain rate as invariant. In the simulation, a developed CA model is built to describe the process of microstructural evolution, and the simulation area is 400 μm × 400 μm.
Figure 8 shows the simulation results of DRX grain sizes at pre-stress of 0, 67, 133, and 167 MPa. From the figure, it can be found that the initial grain size is about 40 μm. When pre-stress increases, a large number of equiaxed fine grains appear on the original grain boundaries, which can be found in Figure 8b–d. The sizes of DRX grains decrease from 40 μm to 25 μm with increasing of pre-stress, since the pre-stress can increase the dislocation density of thermal deformation region, promote DRX nucleation, and then refine grain size. Comparing the simulation diagrams, it can be concluded that the uniformity and numbers of grain size increase with increasing pre-stress. Therefore, appropriately increasing pre-stress is beneficial for the microstructure refinement and uniformity.

### 3.4. Microstructure Observations

After a series of PSHG tests, the optical microstructures of 42CrMo steel were investigated at pre-stress of (0, 67, 100, and 167 MPa) and grinding depth of 200 µm. Figure 9a–d shows the results of metallographic observations under different pre-stress. Experimental results indicated that the higher pre-stress, the finer the grain size, and the calculation results of average grain sizes were 38.6 μm, 35.4 μm, 29.5 μm, and 22.3 μm respectively. Meanwhile, more DRX nuclei were formed at higher pre-stress. In the case of higher pre-stress, it can provide a driving force for boundary movement and store energies for nucleation, and this process is achieved by the movement of large angle grain boundaries. However, the pre-stress is better not loaded too large, or else fold and crack will occur on workpiece surface. It is well-known that the pile up dislocation easily forms a sub-grain boundary, which will produce more nucleus sites for DRX grains. However, the strain will not reach the critical value for complete dynamic recrystallization, so the DRX fraction is relatively small.

As well as pre-stress, the microstructure is also affected by grinding temperature. The results of metallographic observations at different grinding depths of 100, 150, and 250 μm are shown in Figure 9e–f. With grinding temperature increasing, the DRX fraction shows an increasing trend. The average grain sizes were measured as 39.5 μm, 38.2 μm, and 30.2 μm respectively. At a relatively high grinding temperature, usually the DRX grain will grow up, but the results are reversed because the contact time between grinding wheel and workpiece is very transient during PSHG process. Therefore, there is not enough time for DRX grain growth. Instead, DRX fraction will increase under the effect of higher thermal activation energy. It can be seen that the small equiaxed grains take place the original grains and located along the larger grains’ boundaries. However, the effect of grinding temperature on the DRX grain size is not more significant than pre-stress.

The grain sizes under different pre-stresses and grinding depths are summarized in Figure 10. Comparing the different conditions, the grain size is obviously refined when pre-stress is larger than 100 MPa and grinding depth is larger than 200 μm. Meanwhile, it is noticeable that grain size is sensitive to pre-stress, and most of the DRX grains occurred when pre-stress was larger than 100 MPa.

## 4. Conclusions

In this study, the effects of pre-stress and grinding temperature on DRX behavior in 42CrMo steel were investigated though PSHG tests. The kinetic equations of the dynamically recrystallized process and the relationship between pre-stress and grain size were investigated based on a series of experimental and simulated results. The following conclusions were drawn:

(1) In the present paper, the dynamic recrystallization behavior in 42CrMo steel during the PSHG process was investigated though a range of pre-stress and temperatures, and the results show that recrystallization volume fraction increases with the increase of pre-stress and temperature.

(2) The result shows that flow stress follows a power function with DRX grain size, and meanwhile, follows a linear function with pre-stress under a lower strain. A developed CA model under specified grinding conditions is constructed to further verify the correctness of kinetic equations.

(3) DRX grains of 42CrMo steel take place on the boundary of original grains when the deformation temperature and pre-stress are greater than 1223 K and 67 Mpa, respectively. Meanwhile, considering the surface roughness and mechanical properties of the workpiece, the pre-stress should not be too large. Therefore, the completely dynamic recrystallization process will not occur during PSHG tests. 

## Figures and Tables

**Figure 1 materials-12-03124-f001:**
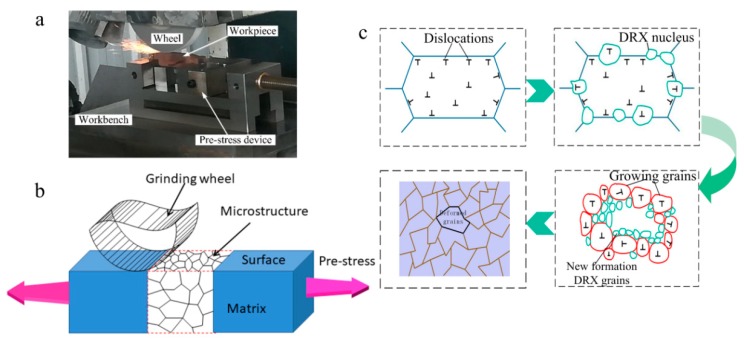
(**a**) The pre-stress hardening grinding (PSHG) experimental setup; (**b**,**c**) Schematic diagram for forming dynamic recrystallization (DRX) grains.

**Figure 2 materials-12-03124-f002:**
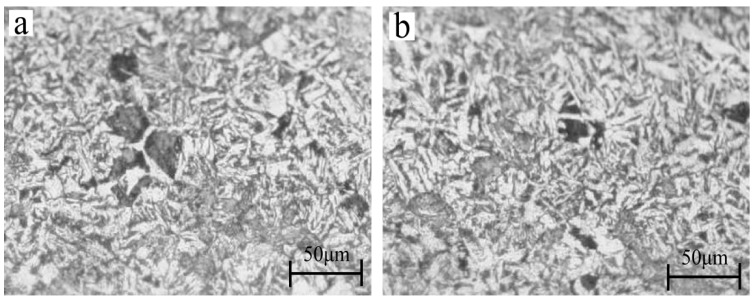
The martensite microstructure under the conditions of (**a**) grinding depth= 200 μm, pre-stress= 67 MPa; (**b**) grinding depth = 200 μm, pre-stress = 100 MPa.

**Figure 3 materials-12-03124-f003:**
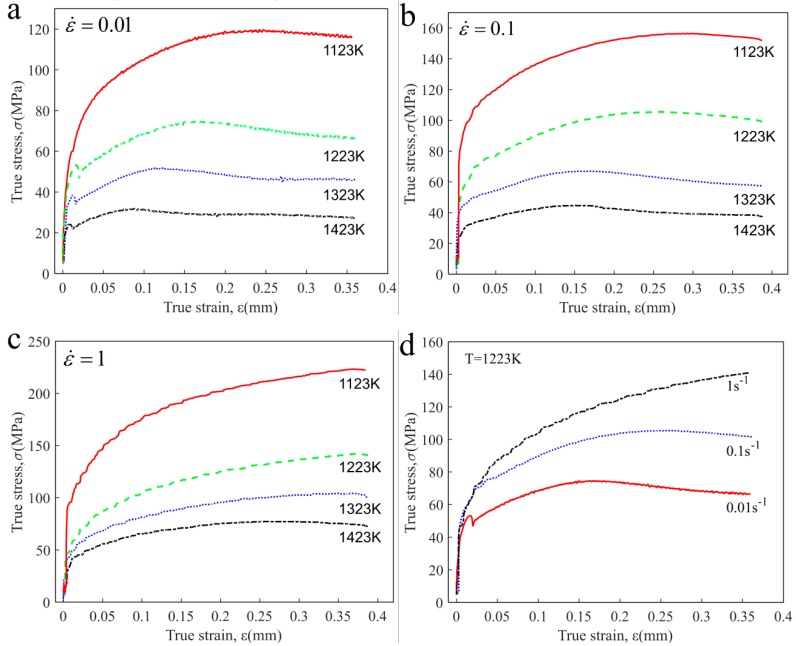
True stress-strain curves of 42CrMo steel at different conditions: (**a**) $\dot{\varepsilon \;}$ = 0.01; (**b**) $\dot{\varepsilon \;}$ = 0.1; (**c**) $\dot{\varepsilon \;}$ = 1; (**d**) T = 1223 K.

**Figure 4 materials-12-03124-f004:**
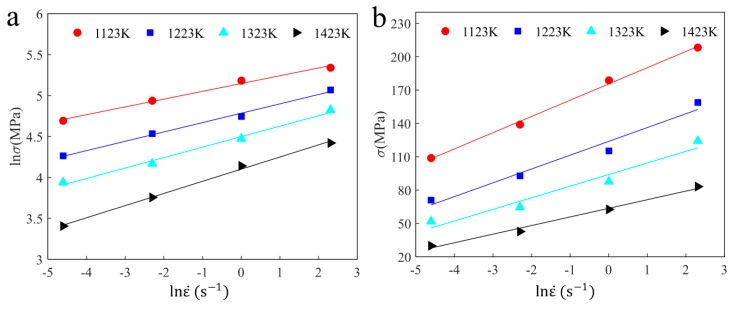
The relationships between (**a**)$\;ln\dot{\varepsilon }-ln\sigma$, (**b**) $\;ln\dot{\varepsilon }-\sigma$.

**Figure 5 materials-12-03124-f005:**
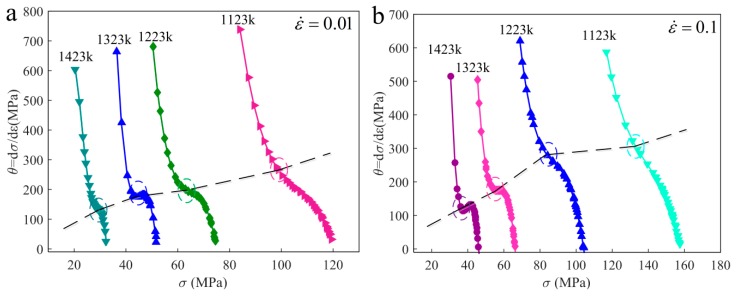
The hardening rate curves under different strain rates (**a**) $\dot{\varepsilon }$ = 0.01 s^−1^; (**b**) $\dot{\varepsilon }$ = 0.1 s^−1^.

**Figure 6 materials-12-03124-f006:**
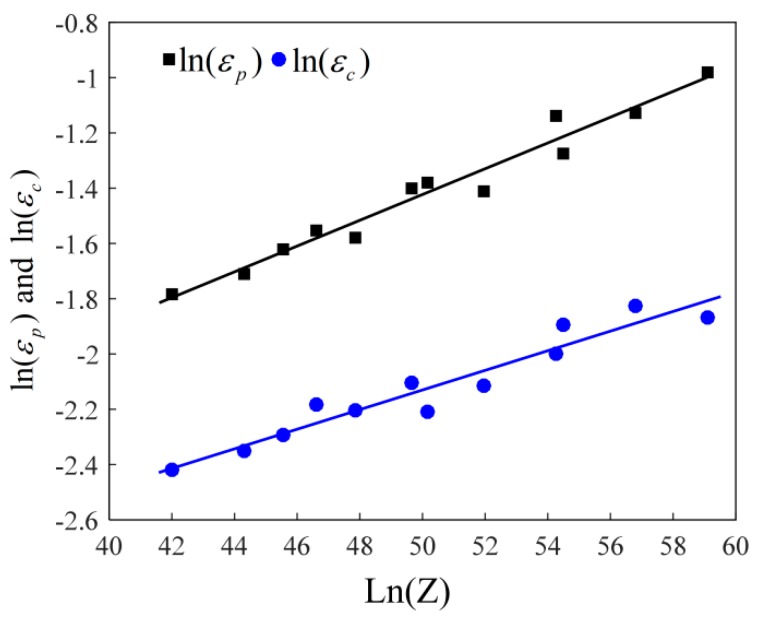
Relationships between ln (*Z*) − ln (*ε_p_*) and ln (*Z*) − ln (*ε_c_*).

**Figure 7 materials-12-03124-f007:**
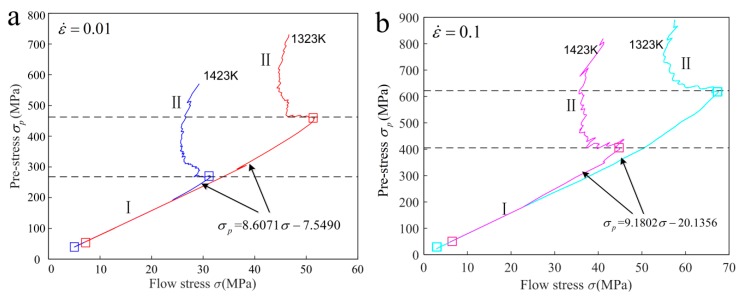
The functional relationship curves (**a**) $\sigma -\sigma_{p}$, strain rate = 0.01 s^−1^; (**b**) $\sigma -\sigma_{p}\;$, strain rate = 0.1 s^−1^; (**c**) $\sigma -d_{\mathrm{drx}}$, strain rate = 0.01 s^−1^; (**c**) $\sigma -d_{\mathrm{drx}}$, strain rate = 0.1 s^−1^.

**Figure 8 materials-12-03124-f008:**
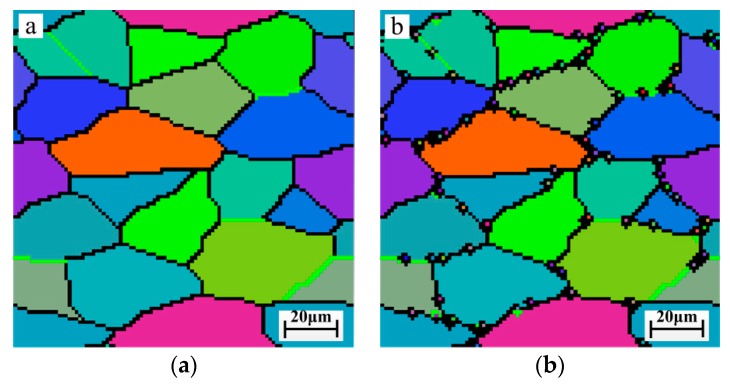
Microstructure evolution model under different pre-stress (**a**) pre-stress $\sigma_{p}\;$ = 0 MPa; (**b**) pre-stress $\sigma_{p}\;$ = 67 MPa; (**c**) pre-stress $\sigma_{p}\;$ = 133 MPa; (**d**) pre-stress $\sigma_{p}\;$ = 167 MPa.

**Figure 9 materials-12-03124-f009:**
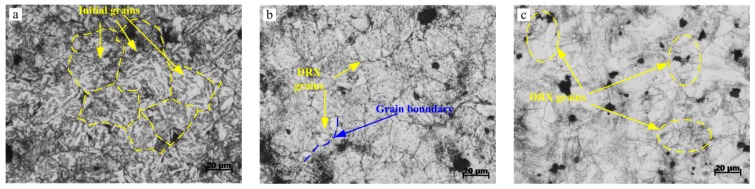
Surface microstructure of 42CrMo steel under different pre-stresses and grinding depths. (**a**) $\sigma_{p}=0\;\mathrm{MPa}$, *α_p_* = 200 μm; (**b**) $\sigma_{p}=67\;\mathrm{MPa}$, *α_p_* = 200 μm; (**c**) $\sigma_{p}=100\;\mathrm{MPa}$, *α_p_* = 200 μm; (**d**) $\sigma_{p}=167\;\mathrm{Mpa}$, *α_p_* = 200 μm; (**e**)$\;\sigma_{p}=67\;\mathrm{MPa}$, *α_p_* = 100 μm; (**f**) $\sigma_{p}=67\;\mathrm{MPa}$, *α_p_* = 150 μm; (**g**)$\;\sigma_{p}=67\;\mathrm{MPa}$, *α_p_* = 250 μm.

**Figure 10 materials-12-03124-f010:**
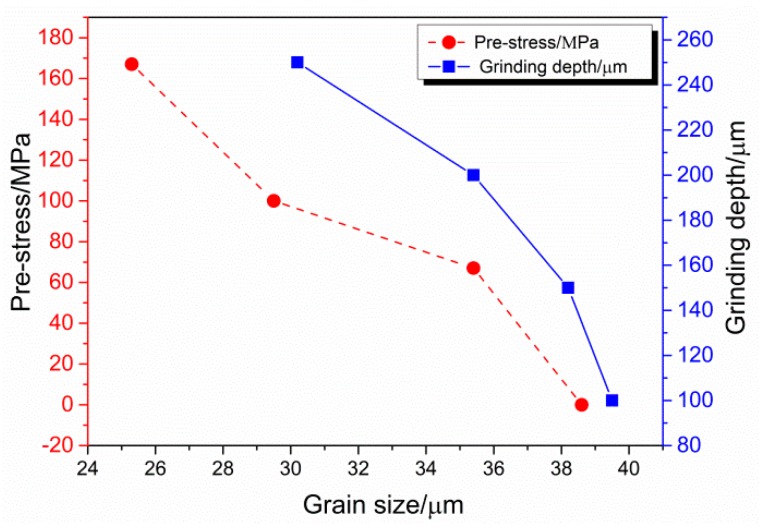
Grain sizes of different pre-stresses and grinding depths.

**Table 1 materials-12-03124-t001:** Composition of 42CrMo steel.

Element	C	Si	Cr	Mn	Mo	P	S	Cu	Fe
**Component%**	0.450	0.280	0.960	0.630	0.190	0.016	0.012	0.014	Balance

**Table 2 materials-12-03124-t002:** Grinding parameters of pre-stress hardening grinding (PSHG) tests.

No.	Feed Rate (*v_w_*)/(mm·s^−1^)	Wheel Speed (*v_s_*)/(m·s^−1^)	Grinding Depth (*α_p_*)/μm	Pre-Stress (*σ*_p_)/MPa
1	20	35	200	0
2			200	33
3			200	67
4			200	100
5			200	133
6			200	167
7			100	67
8			150	67
9			250	67

**Table 3 materials-12-03124-t003:** Values of $\varepsilon_{c}$ and $\varepsilon_{p}\;$ at different deformation conditions.

True Strain.	Strain Rates (s^−1^)	Temperature (K)			
		1123	1223	1323	1423
$\varepsilon_{c}$	0.01	0.1502	0.122	0.10074	0.0889
	0.1	0.1613	0.1210	0.11012	0.0951
	1	0.1545	0.1355	0.11002	0.1129
$\varepsilon_{p}$	0.01	0.2800	0.2464	0.19785	0.1683
	0.1	0.3244	0.2442	0.20661	0.1804
	1	0.3757	0.3198	0.25214	0.2122

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
