# Peer review of "Controlling Grain Sizes of 42CrMo Steel by Pre-Stress Hardening Grinding"

_materials, 2019, doi:10.3390/ma12193124_

Round 1
Reviewer 1 Report
Please find the attached report.

Author Response
Responds to the reviewer’s comments:
Reviewer #1:
1- In the introduction section, the cases of selecting PSHG instead of heat treatment to hardened the part surface and refinement the microstructure of 42CrMo Steel should be illustrated and supported with more references. Also, the applications related to applying PSHG should be stated.
Response: Considering the reviewer’s suggestion, we supplement some sentences in the introduction section to illustrated the advantages of selecting PSHG technology, and the amendments are highlighted in red in the revised manuscript. At present, the technology is still in the stage of theoretical research, but the study is beneficial to perfect the Grind-Hardening theory and prepare for application.
2- Section 2.1; it is recommended to present the chemical composition of 42CrMo Steel and the machine characteristic for PSHG in separate tables. Also, the number of samples should be stated.
Response: Considering the Reviewer’s suggestion, we present the chemical composition of 42CrMo Steel in Table 1. The number of samples have been listed in Table 2.
3- A sub-section from section 2 should illustrate the theory of the developed model.
Response: As Reviewer suggested that we illustrate the theory of the developed model in sub-section 2.3. Revised portion are marked in red from lines 97 to lines 105 in this paper.
4- The developed model should be presented first before being validated with the experimental results.
Response: It is really true as Reviewer suggested that developed model should be presented first before being validated with the experimental results. We have revised some expressions and adjust the order of some paragraphs in the paper. Such as the paragraph in Section 3.1 (from Base on experimental results… to Where σ represents flow stress…) was adjusted to line 120. the expression of some sentences is not accurate enough, for instance, the sentence in line 226 has been revised. At here, I'd like to explain some experimental results are used to calculated parameters of the developed model. So, they are presented before the developed model.
5- The standard deviation and repeatability of the measurements should be discussed on the plots and along with the manuscript text.
Response: Repeatability can be express by standard deviation of single measurement result, which can be expressed as
Because the standard deviation and repeatability is different for each measurement results. So, we don’t know how to discussed on the plots and along with the manuscript text. If reviewer think it is necessary to discuss it, we will try to do that.
6- The trends obtained from the current work should be supported with literature studies to be verified in the discussion section.
Response: The trends between flow stress and pre-stress is the innovate of this study and is obtained from experimental results. Maybe there are some literature studies about this trends, but we didn't find at present. Besides the PSHG is a new technology, there are few literature studies about this field. Therefore, the trends between pre-stress and grain size or grinding depth and grains haven't been studied before. There are some literature studies about the relationship between flow stress and grain size, but the parameters are different from this experiment. So we can't provide literature studies to verify the trends obtained from the current work.
Special thanks to you for your good comments.
Reviewer 2 Report
Dear Authors,
The paper is generally well written, but before publication should be revised. Please consider the points listed below.
Page 1 line 8 – please correct the words “corresponding author” Please add nomenclature – some abbreviations are not explained (!) Table 1 – How was controlled prestress during experiment? Why for 100, 150, 250 (grinding depth) assumed only 67 MPa prestress? Please comment it in details 2. Some parts of image are fuzzy – pleasy try provide as sharp as possible images 3. It will be better to provide results using engineering stress-strain. So, please collect all data for it and give values of – yield/sigma0.2, ultimate tensile strength, elongation at fracture, necking etc…. Model 5 -…How many specimens were used for evaluation of the statistical error? Please provide the confidence levels of the constants…
With best regards
Reviewer
Author Response
Reviewer #2:
1- Page 1 line 8 – please correct the words “corresponding author” Please add nomenclature – some abbreviations are not explained.
Response: As Reviewer suggested that the words “corresponding author“ has been corrected as "Correspondence:orresponding author". We have checked again the paper from beginning to end. The abbreviations such as PSHG and DRX which has been explained in Abstract. The abbreviation of CA which has been explained in section 2.3. The explained portion are marked in red in the paper.
2- Table 1 – How was controlled prestress during experiment? Why for 100, 150, 250 (grinding depth) assumed only 67 MPa prestress? Please comment it in details.
Response: The pre-stress value can be control by applying torque, for example, when the torque were 20, 40, and 60 N·m, the pre-stress were 33, 66, and 100 MPa. The values can be obtained by conversion formula which has been presented in our previous articles published in IJAMT. Therefore, we didn't discuss how to obtain the pre-stress in this study. The relationship between pretightening torque and pre-stress can be expressed as
The reasons why for 100, 150, 200, 250 (grinding depth) assumed only 67 MPa prestress is because of we want to remain pre-stress unchanged to discuss the effect of grinding depth on grain size.
3- Some parts of image are fuzzy – pleasy try provide as sharp as possible images.
Response: As Reviewer suggested that we resubmit all the images again and make them as sharp as possible. The new image resolution all reach or more than 600×600dpi.
4- It will be better to provide results using engineering stress-strain. So, please collect all data for it and give values of – yield/sigma0.2, ultimate tensile strength, elongation at fracture, necking etc.
Response: On this issue, the results of all references quoted in this study use flow stress-strain (true stress-strain). We know it's better to use engineering stress-strain under tension conditions. But we have to consider the surface integrity, especially surface roughness. The degree of deformation is not large, therefore we didn't measure the data of ultimate tensile strength, elongation at fracture, necking etc. On the other hand, the paper mainly focuses on effect of PSHG process on grain size rather than the tensile properties of materials. Can we not provide results using engineering stress-strain?
5- Model 5 -How many specimens were used for evaluation of the statistical error? Please provide the confidence levels of the constants.
Response: Since the average grain sizes were measured using linear intercept method, we select 10 lines on different positions of each specimen surface to measure the average grain sizes. The confidence levels is 90%. For example, the measurement results of grain size for No.4 specimen are 30.4, 29.6, 28.5, 31.6, 29.7, 29.8, 26.4, 32.3, 28.9 and 27.8μm. The standard deviation and error range are 1.734 and 1.005 respectively. The confidence interval are from 28.495μm to 30.505μm. It means that there is 90% probability for the measurement results located at the range from 28.495μm to 30.505μm.
Special thanks to you for your good comments.
Reviewer 3 Report
The present manuscript present a study on controlling of grain sizes by pre-stress hardening grinding. Overall, the manuscript is appropriately organized, the research design is appropriate, the conclusions are supported by results, which were discussed and connected by citation of current work in the area.
Following aspects should to be addressed before publication:
Quality of Figure 2b is not suitable for publication because the greater part is out of focus. All images of Figure 9 should be larger in order to make visible all microstructure features.
Author Response
Reviewer #3:
1-Quality of Figure 2b is not suitable for publication because the greater part is out of focus.
Response: As Reviewer suggested that we have resubmitted Figure 2b in the revised manuscript.
2- All images of Figure 9 should be larger in order to make visible all microstructure features.
Response: The image pixels of all Figure 9 is 1769×1344. It is because of typesetting of images make it a little unclear. We hope this issue decide by editor.
Special thanks to you for your good comments.
Round 2
Reviewer 1 Report
Please find the attached report.

Author Response
Thank you for your comments concerning our manuscript entitled “Controlling grain sizes of 42CrMo steel by pre-stress hardening grinding”. (ID: materials-585098). Those comments are all valuable and very helpful for revising and improving our paper. We have studied comments carefully and have made correction which we hope meet with approval.
Responds to the reviewers’ comments:
1- In the introduction section, regarding the impact of PSHG on the residual stresses along with the processed surface layers, a critical review of the following reference is recommended:
Zhang, J., Wang, G.C. and Pei, H.J., 2017, November. Effects of grinding parameters on residual stress of 42CrMo steel surface layer in Grind-hardening. In International Symposium on Mechanical Engineering and Material Science (ISMEMS 2017). Atlantis Press.
Response: After a careful consideration, we think the reference suggested by reviewer is more appropriate to illustrate the impact of PSHG on the residual stresses along with the processed surface layers. We have added it into our reference.
2- Section 2.3; the theory of the developed model section should present the main parameters and equations to well describing the PSHG process.
Response: As Reviewer suggested that we present the main equation to describe the relationships between flow stress module, thermal activation energy module, recrystallization fraction module and grain size module which is shown in Section 2.3.
3- Higher resolution images are needed to replace the current images in Figure 2.
Response: As Reviewer suggested that we have resubmitted Figure 2 in the revised manuscript. Due to the edge part which out of focus has been removed, so the images resolution maybe reduced than original images.
4- A variety of line styles need to be included in addition to the colour change for the plots in Figures 3-7 to easily access the data by the reader through the black\white copy.
Response: Considering the reviewer’s suggestion, we resubmit Figure 3(a)-3(d) in the revised manuscript. As for other Figures, the data are come from different symbols which can be easy read even though the black\white copy.
5- Figure 8, a short title regarding the shape of the grains before and after PSHG need to be added to the figure.
Response: As Reviewer suggested that we have added a short title under the Figure 8 in the revised manuscript.
6- Some of the expected applications regarding the use of PSHG need to be included in the abstract or conclusion for more attraction to the reader.
Response: At present we have some ideas about the expected applications regarding the use of PSHG which is installed in grinding robot. When the ideas are mature enough we will apply for a patent. But now it is still in the research stage, so I am not sure the specific applications.
Special thanks to you for your good comments.
Reviewer 2 Report
OK - The paper is acceptable!.
Congratulations
Author Response
Special thanks to you for your good comments.